# Enterovirus A71 VP1 Variation A289T Decreases the Central Nervous System Infectivity via Attenuation of Interactions between VP1 and Vimentin In Vitro and In Vivo

**DOI:** 10.3390/v11050467

**Published:** 2019-05-22

**Authors:** Huimin Zhu, Yujuan Cao, Weitao Su, Shan Huang, Weizhi Lu, Yezhen Zhou, Jing Gao, Wei Zhao, Bao Zhang, Xianbo Wu

**Affiliations:** Department of Epidemiology, School of Public Health, Southern Medical University, Guangzhou 510515, China; gg3612@163.com (H.Z.); caoyujuan2090@163.com (Y.C.); swt937872090@163.com (W.S.); 15625985968@163.com (S.H.); pigiraffe@126.com (W.L.); zhyezhen@163.com (Y.Z.); mirrow2@i.smu.edu.cn (J.G.); zhaowei@smu.edu.cn (W.Z.)

**Keywords:** enterovirus A71, VP1 variation, vimentin, central nervous system infectivity

## Abstract

Vimentin (VIM) is a surface receptor for enterovirus-A71, mediating the initial binding and subsequent increase in EV-A71 infectivity. The caspid protein VP1 variation, A289T, is reportedly closely associated with less severe central nervous system (CNS) infections in humans. However, it is unclear whether VIM is associated with a reduction in CNS infections of EV-A71 in the presence of A289T. We investigated whether VIM served as a receptor for EV-A71 in the presence of an A298T substitution in VP1. EV-A71-289A and EV-A71-289T were used to infect human rhabdomyosarcoma cells, control human brain microvascular endothelial cells (HBMECs), and VIM-knockout (KO) HBMECs and inoculated BALB/c mice, SV129 mice, and VIM-KO SV129 mice. Furthermore, we cloned VP1-289A-Flag and VP1-289T-Flag proteins for co-immunoprecipitation analysis. Analysis of viral function revealed that the capacity of viral attachment, replication, and protein synthesis and secretion decreased in HBMECs during an EV-A71-289A infection, the infectivity being higher than that of EV-A71-289T upon VIM-KO. Histopathological and immunohistochemical analyses of brain tissue revealed that cerebral cortical damage was more extensive in EV-A71-289A than in EV-A71-289T infections in control SV129 mice; however, no significant difference was observed upon VIM-KO. Co-immunoprecipitation analysis revealed an interaction between VP1 and VIM, which was attenuated in VP1 harboring A289T; however, this attenuation was reversed by VIM (1-58) peptide. The A289T variation of VP1 specifically decreased the virulence of EV-A71 in HBMECs, and the attenuated interaction between VP1 harboring the A289T variation and VIM essentially decreased the CNS infectivity of EV-A71 in vitro and vivo.

## 1. Introduction

Human enterovirus A71 (EV-A71) is one of the major pathogens that not only causes hand, foot, and mouth disease (HFMD) but also severe central nervous system brainstem encephalitis and aseptic encephalitis [1,2]. These neurological infections are accompanied by a high mortality rate and poor patient prognosis [3,4]. Since its first report in 1974 [5,6], the occurrence of EV-A71-related HFMD has been reported worldwide [7,8,9,10,11], posing serious threats to public health.

EV-A71 infections occur through viral structural proteins including VP1 [12]. Thus far, three receptors for EV-A71 have been reported, including Vimentin (VIM), *p*-selectin glycoprotein ligand 1 (PSGL-1), and scavenger receptor class B member 2 (SCRAB2) [13,14,15]. Among these receptors, VIM is the class-III intermediate filaments found in various non-epithelial, especially mesenchymal cells, glial cells and Human Brain Microvascular Endothelial Cells (HBMECs), responsible for maintaining cell shape, integrity of the cytoplasm and stabilizing cytoskeletal interactions [16,17]. Glial cells and HBMECs are the main components of the Blood Brain Barrier, playing an important role in the process of viral infection of the central nervous system. Several studies have reported that the cell surface VIM plays an important role as a receptor for neurotrophic viruses including Japanese encephalitis virus, Varicella Zoster Virus and Dengue Virus. [18,19,20]. Vim is also reported to be an attachment receptor for EV-A71 [21]. However, the association between VIM and the central nervous system infectivity of EV-A71 is unclear.

EV-A71 is a non-enveloped, single-stranded, positive-sense RNA virus of the family *Picornaviridae*. The open reading frame of EV-A71 encodes four viral capsid proteins (VP1–VP4) and non-structural proteins [22]. Among these proteins, capsid protein VP1 is the most important antigenic determinant of EV-A71 virus and is associated with its virulence [23]. VP1 variations directly influence the virulence of EV-A71. We previously investigated the association between the severity of clinical symptoms and sex, age, viral genotype, and VP1 variations in Guangdong province using the Logistics regression analysis, and reported 48 variations in VP1, with A289T being closely associated with less severe CNS infections in humans. When the 289th amino acid of VP1 was T (threonine), the incidence of severe disease significantly decreased [24]. Further analysis of the spatial structure of VP1 revealed that the 289th amino acid residue was located on the viral surface [25], speculating that the variation A289T altered the interaction between the virus and the neuron-specific receptor VIM.

Thus far, the association between the A289T variant of VP1 and the mechanism underlying EV-A71 infection in the CNS is unclear. This study aimed to analyze neuron-specific protein VIM as a potential receptor for EV-A71 to infect human brain microvascular endothelial cells (HBMECs) and the interaction between the A289T VP1 variant and VIM.

## 2. Materials and Methods

### 2.1. Reagents and Antibodies

Anti-EV-A71 VP1 antibody was obtained from Abnova (Taiwan, China); anti-EV-A71 3c antibody, Abconal (Wuhan, China); anti-vimentin antibody, Santa Cruz (Dallas, Texas, USA); anti-β-actin antibody, anti-mouse and anti-rabbit secondary antibodies and electrochemiluminescent (ECL) Horseradish Peroxidase (HRP) substrate was used for western blotting, (Bioworld, Minneapolis, MN, USA); normal mouse IgG, Sigma (St. Louis, MO, USA); the Co-Immunoprecipitation (Co-IP) Kit, (Thermo Fisher, Rockford, IL, USA); the VIM (1–58) peptide, Ontores Biotech (Hangzhou, Zhejiang, China).

### 2.2. Cell Culture, Virus Isolates, Propagation, and Titer Determination

Human rhabdomyosarcoma (RD) cells, 293T cells, human brain microvascular endothelial cells (HBMECs), control HBMECs, and VIM-knockout (KO) HBMECs were cultured in double-modified Eagle’s medium (DMEM) or 1640 medium supplemented with 10% fetal bovine serum (Gibco, Gaithersburg, MD, USA) at 37 °C and 5% CO_2_.

The full-length infectious clone of EV-A71-289A (C4) was generously provided by Prof. Qin Chengfeng (Academy of Military Medical Science, Beijing, China). An infectious clone of EV-A71-289T (C4) was constructed via site-directed mutagenesis using the Hieff Mut Site-directed Mutagenesis kit (Yeasen Biotech Co., Ltd., Shanghai, China). These two infectious clones were both transfected into RD cells, and the culture supernatant was harvested upon observation of cytopathic effects (CPEs). Two types of viruses were propagated in RD cells and stored at −80 °C [26,27]. To determine the viral titer, serially diluted viral suspensions were inoculated in RD cells in 96-well plates. Upon observation of CPEs, the titers were determined via the Reed–Muench calculation method.

### 2.3. Real-Time Quantitative PCR

Cells were infected at different values of multiplicity of infection (MOI) for 1 h at 37 °C. To detect the level of intracellular viral RNA, at several timepoints after infection, total RNA was extracted using Trizol (Takara, Dalian, China) in accordance with the manufacturer’s instructions. Thereafter, 1 μg of total RNA was reverse-transcribed to cDNA, using the PrimeScript RT reagent kit with gDNA eraser (TAKARA, Dalian, China), and real-time quantitative PCR was performed using BestarTM qPCR MasterMix (DBI® Bioscience, Ludwigshafen, Germany) with EV-A71 sense primer 5-′CCAGAAGAATTTTACCATGAAGTTGT-3′ and antisense primer 5′-AGGGCTCTGCTCATACTATC-3′ and probe-CAGACGGGCACTATACAGGGAG under the following conditions: 1 cycle at 95 °C for 30 s, followed by a three-step procedure consisting of 15 s at 95 °C, 15 s at 55 °C, and 20 s at 72 °C for 40 cycles (with data collection at the end of the 72 °C step at each cycle), and cooling at 37 °C for 30 s. mRNA expression levels were normalized to those of *GAPDH* under the following conditions: 1 cycle at 95 °C for 30 s, followed by a three-step procedure consisting of 15 s at 95 °C, 15 s at 55 °C, and 15 s at 72 °C for 40 cycles (with data collection at the end of the 72 °C step at each cycle), and melting procedure consisting of 10 s at 95 °C, 60 s at 65 °C, and 1 s at 97 °C and cooling at 37 °C for 30 s. mRNA expression levels were normalized using the 2^−ΔΔCt^ method and the results were expressed as an expression fold-change. To detect the level of extracellular viral RNA, at several timepoints after infection, cell supernatant was collected for viral RNA extraction with a QIAamp Viral RNA Kit (QIAGEN, Dusseldorf, Germany). Thereafter, viral RNA was transcribed to cDNA with PrimeScript RT reagent kit with gDNA eraser (TAKARA, Dalian, China). Real-l-time quantitative PCR was performed using BestarTM qPCR MasterMix (DBI® Bioscience, Ludwigshafen, Germany) with EV-A71 primers and probe. The infectious full-length infectious clone of EV-A71 was used as template standards and the level of viral RNA was absolutely quantified. In viral attachment experiments, cells were infected with viruses at different MOI for 1 h at 4 °C. Then, the total RNA was extracted with Trizol (Takara, Dalian, China), transcribed to cDNA with the PrimeScript RT reagent kit with gDNA eraser (TAKARA, Dalian, China) and quantified using the 2^−ΔΔCt^ method.

### 2.4. Western Blot

At several timepoints after infection, cells were scraped off and lysed on ice for 30 min in lysis buffer (Genshare, Xi’an, China). Thereafter, cell lysates were centrifuged at 15,000× *g* for 30 min at 4 °C and the supernatant was harvested. Equal amounts of protein, which was quantitated with Bioepitope Bicinchoninic Acid Protein Assay Kit (Bioworld, Minneapolis, MN, USA) was separated via Sodium Dodecyl Sulfonate-Polyacrylamide Gel Electrophoresis (SDS-PAGE). Thereafter, proteins were electroblotted onto a polyvinylidene difluoride (PVDF) membrane (Bio-rad, Shanghai, China). The membranes were incubated in blocking buffer (3% BSA) for 2 h at room temperature and then probed with anti-EV-A71 3c antibody and kept overnight with gentle agitation at 4 °C, followed by probing with corresponding secondary antibodies for 1 h at room temperature and with ECL reagent for exposure.

### 2.5. Plasmids and Protein Expressing Vector

To construct VP1-289A-Flag and VP1-289T-Flag proteins, the VP1 gene was amplified from the infectious clone, using the VP1 sense primer 5′-CCGGAATTCGCCACCATGGGAGATAGGGTGGCAGATGT-3′ and antisense primer 5′-CGCGGATCCCCGCTACCGCCGCTTCCCTTGTCATCGTCGTCCTTGTAATC GAGAGTGGTGATTGCTGTACG-3. Thereafter, gel purification was carried out, and the gene was cloned into the corresponding restriction sites of pEGFP-N1.

### 2.6. Co-Immunoprecipitation Analysis

Before plasmid transfection, anti-VIM antibody was immobilized on the resin in accordance with the manufacturer’s instructions. At 48 h after plasmids were transfected into 293T cells, using Lipofectamine 3000 Transfection Reagent (Thermo Fisher, Rockford, IL, USA), cell lysates were harvested and pre-cleared in accordance with the manufacturer’s instructions. Thereafter, equal amounts of cell lysates were added to the immobilized antibody and incubated overnight with gentle agitation at 4 °C. The following day, the resins were washed twice and heated in 2× sample buffer. After brief centrifugation, the supernatant was applied for western blotting.

### 2.7. Mice Infection and Organ Collection

All animal protocols were approved by the Southern Medical University Experimental Animal Ethics Committee (Approval Number: L2018018, Approval date: 24 January 2018). The individual ventilated cages (IVC) system was used to offer a suitable environment for mice. BABL/c mice obtained from the Experimental Animal Centre, Southern Medical University, Guangzhou, China, control SV129 mice, and VIM-KO SV129 mice (generously provided by Prof. Huang Shenghe) aged 1 week were intraperitoneally inoculated with 10^8^ and 10^7^ pfu/mL of EV-A71A and EV-A71T. Control mice were inoculated with 1640 medium. Thereafter, clinical symptoms including weakness, reduced activity, tremor, hunchback, ruffled fur, and limb paralysis were monitored [28,29,30]. Mice were euthanized after 10 d of inoculation. After perfusion with PBS, brain tissue and the cerebrospinal fluid were harvested for histopathological and immunohistochemical analyses and to determine the 50% tissue culture infective dose (TCID50).

### 2.8. Statistics Analysis

Data from at least three independent experiments were presented as mean ± SEM using SPSS 19.0 statistical software. The mean optical density (MOD) of staining intensity in immunohistochemical analyses was measured using Image Pro Plus 6.0 software. The difference among treatment groups was analyzed by a Student’s *t*-test. *p* values < 0.05, from a two-tailed test was considered statistically significant.

## 3. Results

To investigate whether the A298T mutation in VP1 affected non-CNS infectivity, the viral kinetics including viral attachment capacity, replication, protein synthesis, and secretion of two strains were detected in RD cells. The level of intracellular and extracellular viral RNA was detected via RT-qPCR at each timepoint after infection of EV-A71-289A and EV-A71-289T in RD cells at a MOI of 5. Intracellular and extracellular viral RNA levels of EV-A71-289A and EV-A71-289T infection had the same tendency at each timepoint after infection (*p* > 0.05) (Figure 1A,B). Furthermore, intracellular viral protein 3c was detected after 12, 24, and 48 h of infection of EV-A71-289A and EV-A71-289T at a MOI of 10. The EV-A71 3c protein was detected and viral protein levels remained constant at EV-A71-289A and EV-A71-289T infection (Figure 1D). A virus binding experiment, wherein RD cells were infected at 4 °C with EV-A71-289A and EV-A71-289T at MOI_S_ of 50, 500, and 5000, was carried out to assess viral attachment to cells, revealing that the attachment of the two viral strains to RD cells increased with an increase in MOI; however, no significant difference was observed in viral attachment capacity in the two strains (*p* > 0.05) (Figure 1C). The viral attachment, replication, protein synthesis, and secretion of two strains was the same in RD cells.

To further investigate whether the A298T variation in VP1 protein affected CNS infectivity through VIM, control HBMECs and VIM-KO HBMECs were infected with EV-A71-289A and EV-A71-289T. Detection of viral infection revealed that intracellular viral RNA of EV-A71-289A infection was 1.3- and 3-fold that of EV-A71-289T infection in control HBMEC at 24 h and 48 h after infection, respectively, while the level of intracellular viral RNA of EV-A71-289A infection was 1.7- and 1.8-fold that in EV-A71-289T infection in VIM knocked-out HBMECs (*p* < 0.05) (Figure 2A). Extracellular viral RNA was quantified to analyze viral secretion. Viral RNA in the EV-A71-289A infection was 2.2- and 1.6-fold that in the EV-A71-289T infection in control HBMECs at 24 and 48 h after infection, respectively, while the level of extracellular viral RNA in the EV-A71-289A infection was 1.4- and 1.2-fold that in EV-A71-289T infection in VIM-KO HBMECs (*p* < 0.05) (Figure 2B). The EV-A71 3c protein was detected after infection at a MOI of 10. These results showed that while viral protein of EV-A71-289A infection was more than that of EV-A71-289T infection in control HBMEC, the EV-A71 3c protein level in the EV-A71-289T infection was greater than that in the EV-A71-289A infection in VIM knocked-out HBMECs (Figure 2D). To determine the viral attachment capacity of the two strains, control and VIM-KO HBMECs were infected at 4 °C with EV-A71-289A and EV-A71-289T, and viral RNA expression levels were quantified via RT-qPCR. At a MOI of 5, levels of viral RNA in the EV-A71-289A infection were 1.3-fold those in EV-A71-289T infection in control HBMECs, while those in the EV-A71-289A infection were 1.2-fold those in the EV-A71-289T infection in VIM-KO HBMECs. Moreover, in both EV-A71-289A and EV-A71-289T infections, viral RNA levels in control HBMECs were greater than those in VIM-KO HBMECs (Figure 2C). These results indicated that VIM was the receptor for EV-A71 in HBMECs, since VIM-KO in HBMECs decreased viral attachment, replication, protein synthesis, and secretion, and the A289T mutation of VP1 decreased the capacity of viral attachment, replication, protein synthesis, and secretion in HBMECs.

To determine whether EV-A71 could directly interact with VIM and whether there were differences in the interaction with EV-A71-289A and EV-A71-289T, viral particles were preincubated with VIM peptide or 1% DMSO for 30 min at 4 °C before infection. At 1 h after infection, total RNA was extracted, and viral RNA was quantified. In control HBMECs, viral RNA levels of both strains decreased when viral particles were preincubated with VIM peptide (1–58), suggesting that EV-A71 could directly interact with VIM. Moreover, in VIM-KO HBMECs, the viral level in infection of two strains had no difference when viral particles were preincubated with VIM peptide. (Figure 3A). The interaction was further verified via co-immunoprecipitation analysis. Cell lysates were harvested and incubated overnight at 4 °C with immobilized anti-Flag antibody or normal IgG after the transfection of expression vectors expressing VP1(289A)-Flag and VP1(289T)-Flag. A 55-kDa band corresponding to VIM was pulled down and detected. Nearly 1.6-fold VIM was pulled down by VP1(289A)-Flag compared with that by VP1(289T)-Flag, while more VIM was pulled down by VP1(289T)-Flag upon addition of VIM peptide and competitive inhibition of the interaction between VP1 and VIM (Figure 3B,C). These results suggest a stronger interaction between VIM and VP1(289A) than with VP1(289T).

After showing that VIM was a receptor for EV-A71 and that the A289T mutation of VP1 decreased the capacity for viral attachment, replication, protein synthesis, and secretion in vitro, we investigated these effects in vivo. BALB/c mice, control SV129 mice, and VIM-KO SV129 mice aged 1 week were intraperitoneally inoculated with 10^8^ and 10^7^ pfu/mL of EV-A71-289A and EV-A71-289T and then monitored. The onset of infection including weakness, reduced activity, tremor, hunchback, ruffled fur, and hind limb paralysis was observed 3 to 7 d after EV-A71-289A infection and 4 to 8 d after EV-A71-289T infection (Figure 4A). Thereafter, the body weight of the mice decreased, even apparently decreased at 10^8^ pfu/mL, in both BABL/c mice and SV129 mice. At the same viral titers, control SV129 mice had a greater body weight than VIM-KO SV129 mice (Figure 4B-a,b). Furthermore, the morbidity rate was higher among EV-A71-289A-infected mice than among EV-A71-289T-infected mice in both BALB/c and SV129 mice. Moreover, in both EV-A71-289A and EV-A71-289T infections, the morbidity rate of VIM-KO SV129 mice was markedly lesser than that of control SV129 mice (*p* < 0.05). These results indicated that VIM-KO could reduce the morbidity rate of EV-A71 infections and that the A289T mutation of VP1 could decrease virulence in vivo (Figure 4B-c,d). To further determine the effect of the A289T mutation of VP1 protein on CNS infectivity, cerebrospinal fluid was harvested for TCID50 analysis, and brain tissue was harvested for histopathological and immunohistochemical analyses. TCID50 analysis of cerebrospinal fluid revealed that when inoculated with 108 pfu/mL, the viral load in cerebrospinal fluid of EV-A71-289A-infected mice was 9.5-fold that of EV-A71-289T-infected mice (*p* < 0.05), indicating that EV-A71-289A was more neurotrophic than EV-A71-289T. This indication was further determined via histopathological and immunohistochemical analyses. In BALB/c mice and SV129 mice, histopathological analysis revealed more extensive neuronal damage and lymphocyte infiltration in the cerebral cortex in EV-A71-289A-infected mice. In particular, among SV129 mice, neuronal damage and lymphocyte infiltration in the cerebral cortex was less extensive in VIM-KO SV129 mice (Figure 5A,C). Regarding immunohistochemical analysis, EV-A71 was detected in the cerebral cortex, and the staining intensity was stronger in BALB/c mice infected at 10^8^ pfu/mL of EV-A71-289A (Figure 5A). Among SV129 mice, the staining intensity was stronger among EV-A71-289A-infected mice at 108 pfu/mL. Furthermore, immunohistochemical staining of the cerebral cortex was widespread in control SV129 mice rather than in VIM-KO SV129 mice (Figure 5D). These results together indicated that the A289T VP1 mutation decreased its neuropathogenesis by attenuating its interaction with VIM in vivo.

## 4. Discussion

Variation in viral proteins is often related to virulence of EV-A71. The N69D variation in the 3D protein structure decreased the virulence of EV-A71 [31]. The 145 variation of VP1 protein and the 149 variation of VP2 protein is essential for attenuating the virulence of EV-A71 [32]. The present study further explored the interaction between the variation A289T of VP1 and the VIM receptor on the basis of our previous molecular epidemiological study reporting that variation A289T was closely associated with less severe CNS infection in humans.

The A289T variation of VP1 specifically decreased its virulence in HBMECs. Basic characteristics of EV-A71-289A and EV-A71-289T were observed in RD cells and HBMECs. The capacity of EV-A71-289T for viral attachment, replication, protein synthesis, and secretion decreased in HBMECs compared with EV-A71-289A, while there was no difference about the capacity in RD cells between the virus of EV-A71-289T and EV-A71-289A. As HBMECs are the primary components of the blood-brain barrier [33,34], this indicated that EV-A71 could penetrate the blood-brain barrier and cause CNS infections through HBMECs. Decreased virulence of EV-A71-289T to HBMECs implied that EV-A71-289T could penetrate the BBB and cause a CNS infection to a lesser extent, concurrent with previous molecular epidemiological data. Hence, a more extensive CNS infection in the cerebral cortex was observed in EV-A71-289A-infected mice. VP1 is a critical structural protein mediating the interaction between EV-A71 and the receptor, and variations in VP1 often alter the interaction between the virus and its receptor [35]. Chen et al. reported that a variation at residue 172 of VP1 promoted EV-A71 binding to SCRAB2 via a canyon of VP1. Decreased virulence in neural tissue has also been reported in amino acid variations at VP1-145 [36]. The present study showed that the A289T variation of VP1, a more significant variation based on the epidemiological study, decreased the CNS infectivity of EV-A71 through decreased penetrability through the blood-brain barrier in vitro and vivo.

Attenuation of the interaction between the A289T VP1 variant and VIM played a critical role in decreasing CNS infection of EV-A71. Upon VIM-KO, viral attachment and replication decreased in HBMECs. Morbidity and the cerebral cortical damage in VIM-KO SV129 mice reduced, indicating that VIM was required for CNS infection of EV-A71. Moreover, the decreased range of viral attachment, replication, protein synthesis, and secretion in the EV-A71-289A infection was larger than that in the EV-A71-289T infection upon VIM-KO. In addition, in VIM-KO SV129 mice, a reduction in the morbidity range in EV-A71-289T-infected mice was smaller than that in EV-A71-289T-infected mice. Moreover, while cerebral cortical damage in the EV-A71-289A infection was more extensive than that in the EV-A71-289T infection or in control SV129 mice, a significant difference was not observed upon VIM-KO. In summary, the A289T decreased viral attachment, replication, and CNS infection but not under VIM-KO, indicating that the reduced interaction between the A289T variant of VP1 and VIM decreased CNS infectivity of EV-A71. VIM is reportedly an attachment receptor for EV-A71 and may mediate the initial interaction, subsequently increasing the infectivity of EV-A71 [21]. In astrocytes, EV-A71 VP1 can activate CaMK-Ⅱ and phosphorylate and rearrange VIM, thereby potentially promoting viral replication [15]. Activation of inflammasomes by VIM was also observed in mice with an EV-A71 infection [37]. The present study not only indicated VIM as a receptor for EV-A71, but also showed that the EV-A71 A289T variant of VP1 decreased the CNS infectivity of EV-A71 owing to attenuated interaction between VP1 and VIM in vitro and vivo. We speculate that attenuated interaction between VP1(289T) and VIM decreased the phosphorylation and rearrangement of VIM, thus decreasing viral replication.

Furthermore, VP1 interacted with the head of VIM through its tail. Co-immunoprecipitation analysis verified the interaction between VP1 and VIM, with this interaction being attenuated in the A289T variant of VP1. This attenuated interaction could be reverted by VIM (1–58) peptide. Analysis of the effect of VIM (1–58) peptide on viral attachment to HBMECs seemingly displayed the same tendency, indicating that the head of VIM was critical in the interaction between VP1 and VIM. As residue 289 was located in the tail of VP1, we speculated that VP1 interacted with the head of VIM through its tail, and the A289T variation attenuated this interaction.

The present study extensively investigated the association between the A289T variation of VP1 in EV-A71 and CNS infection and showed that the EV-A71 VP1 variant A289T decreased CNS infectivity of the virus via attenuation of the interaction between VP1 and VIM in vitro and vivo. Combined with our previous molecular epidemiological investigation in Guangdong Province, we speculate that EV-A71 VP1 variant A289T decreases CNS infectivity of the virus via attenuation of the interaction between VP1 and VIM. Thus, further studies from a monkey model and clinical data are required to clarify the association between EV-A71 VP1 variant A289T and less severe CNS infectivity for predicting the risk of severe EV-A71 infections. Further bioinformatics analysis for predicting the site of interaction for VP1 and VIM is needed. Moreover, this study reported that the head functional region of VIM was a hotspot for polymorphisms in the study population. Therefore, future studies are required to analyze the diversity of VIM and its interaction with VP1 variants on EV-A71 causing CNS infections.

## Figures and Tables

**Figure 1 viruses-11-00467-f001:**
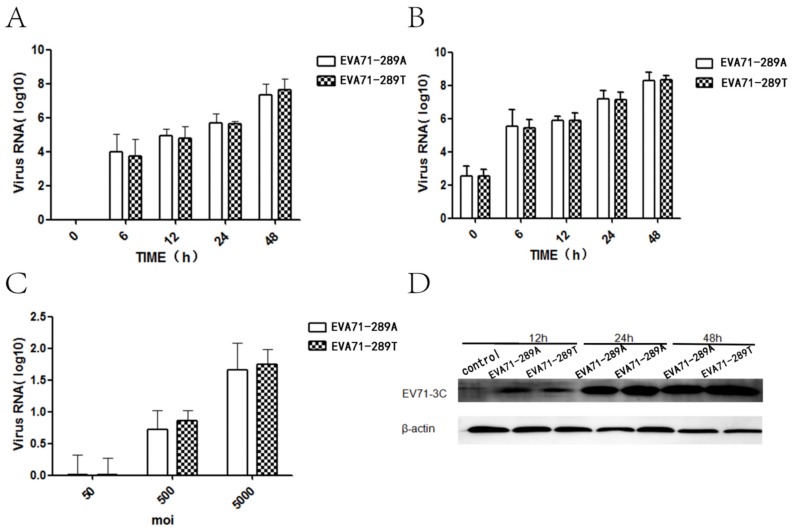
Viral Kinetics of EV-A71-289A and EV-A71-289T in Rhabdomyosarcoma (RD) cells. RD cells were infected with EV-A71-289A and EV-A71-289T. At each time point after infection, intracellular (**A**) and extracellular (**B**) viral RNA expression levels were detected via quantitative RT-qPCR analysis. The capacity of attachment of EV-A71-289A and EV-A71-289T to RD cells was detected via RT-qPCR analysis after infection at 4 °C for 1 h at various values of multiplicity of infection (**C**). Viral protein levels were detected via western blotting after RD cells were infected with EV-A71-289A and EV-A71-289T at each timepoint at a MOI of 5 (**D**).

**Figure 2 viruses-11-00467-f002:**
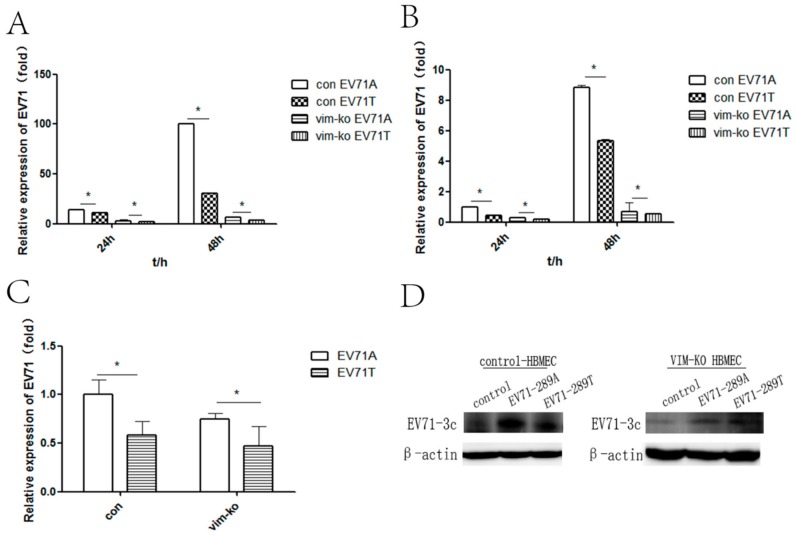
Viral Kinetics of EV-A71-289A and EV-A71-289T in control human brain microvascular endothelial cells (HBMECs) and VIM-KO HBMECs. Control HBMECs and VIM-KO HBMECs were infected with EV-A71-289A and EV-A71-289T. At each timepoint after infection, the levels of the intracellular (**A**) and extracellular (**B**) viral RNA were detected via quantitative RT-PCR (qRT-PCR) analysis. The capacity of attachment of EV-A71-289A and EV-A71-289T to HBMECs was detected via qRT-PCR analysis after viral infections at 4 °C for 1 h at a MOI of 5 (**C**). Viral protein levels were detected via western blotting after HBMECs were infected with EV-A71-289A and EV-A71-289T at a MOI of 10 (**D**). Asterisks indicate significant differences (*p* < 0.05 by *t*-test).

**Figure 3 viruses-11-00467-f003:**
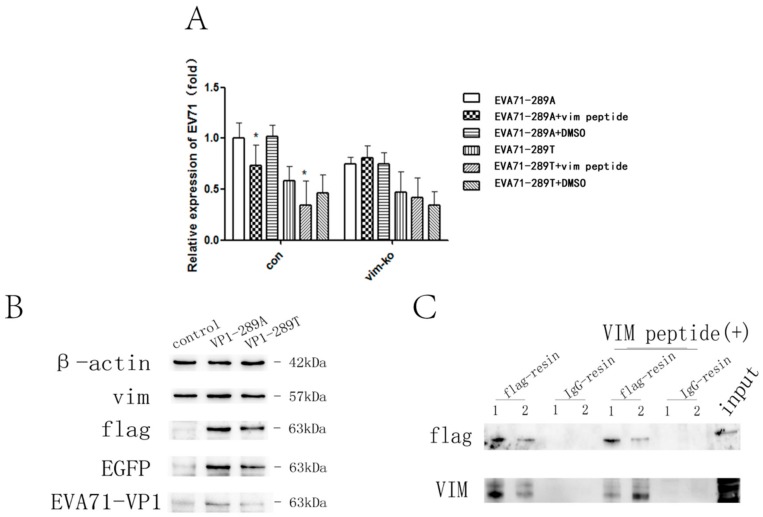
Assessment of the interaction among EV-A71-289A, EV-A71-289T, and VIM. Viral attachment to control human brain microvascular endothelial cells (HBMECs) and VIM-KO HBMECs was detected via quantitative RT-qPCR analysis after infection with EV-A71-289A and EV-A71-289T at 4 °C for 1 h after viral incubation with VIM peptide or DMSO (1%) (**A**). Cell lysates were harvested for western blotting (**B**) and then incubated overnight at 4 °C with immobilized anti-flag antibody or normal IgG after 293T cells were transfected with expression vectors expressing VP1(289A)-flag and VP1(289T)-flag. Figure C shows a western blot of the following precipitated protein samples using anti-Flag and anti-VIM antibodies: VP1(289A)-flag (1) and VP1(289T)-flag (2) incubated with anti-flag-resin; VP1(289A)-flag (1) and VP1(289T)-flag (2) incubated with normal IgG-resin; VP1(289A)-flag (1) and VP1(289T)-flag (2) incubated with anti-flag-resin and VIM peptide; and VP1(289A)-flag (1) and VP1(289T)-flag (2) incubated with normal IgG-resin and VIM peptide (**C**). Asterisks indicate significant differences (*p* < 0.05 by *t*-test).

**Figure 4 viruses-11-00467-f004:**
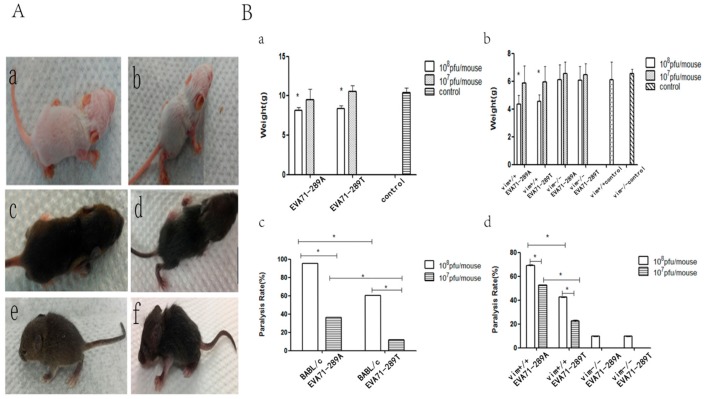
Incidence after EV-A71-289A and EV-A71-289T infections in BALB/c mice and SV129 mice. Onset of infection including weakness, reduced activity, tremor, hunchback, ruffled fur, and hind limb paralysis was observed 3 to 7 d after EV-A71-289A infection and 4 to 8 d after EV-A71-289T infection. (**a**,**c**: Control BALB/c and SV129 mice, **b**,**d**,**e**,**f**: Mice infected by EV-A71). (**A**) Change in body weight and morbidity (**a**,**b**: The change in weight in BALB/c and SV129 mice, **c**,**d**: Morbidity of BALB/c and SV129 mice)(**B**). Asterisks indicate significant differences (*p* < 0.05 by *t*-test).

**Figure 5 viruses-11-00467-f005:**
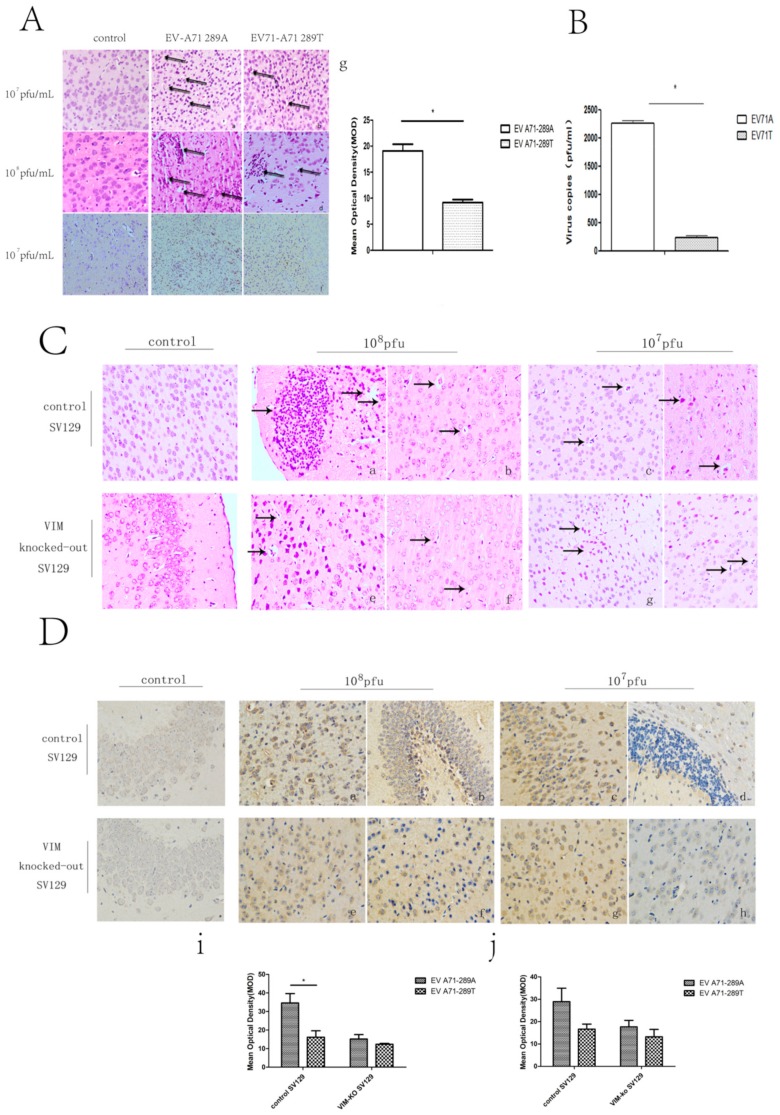
Brain damage detected via histopathological and immunohistochemical analyses in BALB/c mice and SV129 mice after infection with EV-A71-289A and EV-A71-289T. Brain damage detected via histopathological and immunohistochemical analyses in BALB/c mice. (**A**) a, b, c, d: Histopathological analysis of BALB/c mice infected by EV-A71-289A and EV-A71-289T at 10^7^ pfu/mL. The arrows indicate neuronal damage or lymphocyte infiltration. e, f: Immunohistochemical analysis of BALB/c mice infected by EV-A71-289A and EV-A71-289T at 10^8^ pfu/mL. The arrows indicate the positive staining of EV-A71; g: Denote the Mean of Optical Density (MOD) of staining intensity of IHC analysis in BALB/c mice infected by EV-A71-289A and EV-A71-289T at 10^8^ pfu/mL. The 50% tissue culture infectivity dose (TCID50) value of cerebrospinal fluid of BALB/c mice intraperitoneally inoculated with 10^8^ pfu/mL of the virus (**B**). Brain damage detected via histopathological analysis in SV129 mice. (**C**) a,c,e,g denote mice intraperitoneally inoculated with EV-A71-289A; b,d,f,h denote mice intraperitoneally inoculated with EV-A71-289T. Brain damage detected via immunohistochemical analysis in SV129 mice. (**D**) a,c,e,g denote mice intraperitoneally inoculated with EV-A71-289A; b,d,f,h denote mice intraperitoneally inoculated with, EV-A71-289T; i denotes the Mean of Optical Density (MOD) of staining intensity of IHC analysis in SV129 mice infected by EV-A71-289A and EV-A71-289T at 10^8^ pfu/mL; j denotes the Mean of Optical Density (MOD) of staining intensity of IHC analysis in SV129 mice infected by EV-A71-289A and EV-A71-289T at 10^7^ pfu/mL. Asterisks indicate significant differences (*p* < 0.05 by *t*-test).

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
