# Peer review of "Enterovirus A71 VP1 Variation A289T Decreases the Central Nervous System Infectivity via Attenuation of Interactions between VP1 and Vimentin In Vitro and In Vivo"

_viruses, 2019, doi:10.3390/v11050467_

Round 1
Reviewer 1 Report
Zhu et al present an interesting manuscript on the relation between severity of disease with a specific VP1 variation and the subsequent receptor use of EV-A71, an important pathogen for CNS infections. It is often very difficult to relate clinical outcome to one or more specific virological features, and also in this manuscript this is not convincingly shown.
-The abstract starts with a statement on vimentin as the specific receptor for EV-A71 in neurons, while later in the manuscript the authors do experiments to prove that VIM is a receptor for EV-A71. This is confusing and it stays confusing throughout the manuscript. Please start with a comprehensive Introduction where the role of VIM is more clearly explained with more background to indicate what is known already. Now it is presented as being one of the main receptors, while VIM has only been indicated as an attachment receptor, and has been linked to neuropathogenicity, also for JE; the exact role of VIM is much less clear than for the other two main EV-A71 receptors, but this is not clear from the abstract and Introduction.
-minor comment to the Introduction: the referencing goes to quite old literature, which is not wrong, but there is so much very much updated literature for the cited topics; please find a better balance between citing the original work (e.g. ref 5), while for the epidemiology and clinical outcomes more updated references are available.
-the work is largely based on a previous epidemiological study (ref 20), however, I could not find this manuscript, not in Pubmed nor online anywhere else. This was surprising, since that work is the basis for the mutation studied. Unfortunately this was therefore impossible to go back to the original reference.
-The way the authors use the terminology for the VP1 mutation is confusing. In the abstract it is suggested that A289T is associated with severe disease, then in line 51 the T is associated with less severity, then in line 265 again as stated in the abstract, with severe disease. Please be consistent. The VP1-289T is apparently the one that is associated with less severe CNS disease, while the VP1-289A is associated with severe CNS disease.
-The Materials and Methods does not explain how viral titers intra- versus extracellularly are measured. Also, in the Results there is data on attachment, but this is not properly explained in the methods how this is measured.
- In line 146/7 the differences in replication capacity on RD cells are directly to CNS infectivity, this is a major assumption and cannot be made like that. Please delete.
-The figures are quite small, therefore the differences are hard to read.
_Since the Figures are so small, the error bars are difficult to read. How many times were the experiments repeated? There is no mentioning about repeats or duplos intra- and inter assay.
-The legends of Fig 1 and 2 are nearly identical and seem to have the same mistake as well: The (C) is located wrong and (D) is missing.
- After showing that the EV-A71 variants show differences in in vitro cell lines and that VIM is important for entry of human endotheial cells, whre VP1-289A seems to have an advantage, confirmation is sought in infection of mice with the different virus strains. Indeed, in these mice models, the 289A variant is more neurotropic. What the authors fail to acknowledge is that this is only an confirmation of the in vitro data, nothing more. In line 285, the results of the study are directly translated to CNS infection of EV-A71. Of mice? Of men? Transaltion of data from animal models to human pathogenesis is not 1:1. Please downtune o the clinical relevace of these findings in mice. Data from mice, monkeys and humans for the virulence of the 145 variant of VP1 have been different, and this should be referenced in the discussion as an example that more research is needed before conclusions of the relevance of the 289 variant for infection of the human CNS can be made.
As a general remark: many authors are now using EV-A71, instead of EV71. There is no official consensus, but just to consider for the authors whether they want to change it or not.
Author Response
We are very thankful for reviewers’ comments on our manuscript entitled "Enterovirus 71 VP1 variation A289T decreases the central nervous system infectivity via attenuation of interactions between VP1 and VIM" (Manuscript ID: viruses-469602).
The comments from reviewers are all very helpful for improving our paper and meaningful for our research. We have studied all the comments carefully and made revision on our manuscript according to the comments of reviewers. The response to the reviewers’ comments are submitted and we hope it can meet reviewer's approval.

Reviewer 2 Report
no comment to the authors
Author Response
We noticed that reviewer2 didn't comment on our manuscript. So, we revised our manuscript according other 2 reviewers’ comments and we hope that we could recive reviewer's comments on our revised manuscript.
Reviewer 3 Report
The topic and approach are very interesting.
I have a few suggestions
Line 45 page 2 Picornaviridae should be in italics.
The figures are too small, they should be larger.
The legends of figures are not clear especially Figure 5. The legend should spell out the histopathological change in few words, and even the IHC analysis.
Maybe a table of staining intensity should be included – to make clear.
The figures never show comparative staining intensities clearly.
Abbreviations have to be explained not all have been included eg. IVM
The animal ethics committee permissions are not mentioned.
2.7 of materials and methods should be changed to "Mice infection and organ collection".
English should be improved in general especially the Materials and Methods
Author Response
We are very grateful for reviewer's comments on our manuscript entitled "Enterovirus 71 VP1 variation A289T decreases the central nervous system infectivity via attenuation of interactions between VP1 and VIM" (Manuscript ID: viruses-469602).
The comments from reviewers are all very helpful for improving our paper and meaningful for our research. We have studied all the comments carefully and made revision on our manuscript according to the comments of reviewers. The main response to to the reviewer's comments are submitted. Once again, we hope that our revision could meet with approval. Once again, thank you for your comments and suggestions.

Round 2
Reviewer 1 Report
The manuscript has been improved significantly, all my points have been answered properly.
A few minor comments:
line 13, VIM is (the word 'is' is missing here)
line 15, and 167: I would suggest to replace"decreased severe" CNS infections by less severe.
Please also indicate clearly when data are related to studies in men, or mice (or monkeys?). In the Abstract, it is unclear whether the presented data are from men or mice studies. For example in line 61, add 'in humans' when it applies to human studies.
I tried to look for ref 22, still could not find it. Please add the doi to the reference in the manuscript so that other readers might be able to find it.
Line 167 is still unclear. I believe the authors aim at showing that replication kinetics and infectivity are the same in RD cells for both mutants. I would recommend to change the introductionary sentence in order to make that more clear.
Author Response
Dear Reviewer,
Thank you very much for your comments and suggestions. We have revised the manuscript according to the comments and suggestions and the revisions were highlighted using the 'Track Changes' function. We response to the comments point by point in the uploaded Word file.
Once again, thank you very much for your comments and suggestions.
Your sincerely
